# Hydrogel oxygen reservoirs increase functional integration of neural stem cell grafts by meeting metabolic demands

Y. Wang [1,2,3], E. R. Zoneff[1,3], J. W. Thomas[1], N. Hong [4], L. L. Tan [4], D. J. McGillivray [5], A. W. Perriman[4,6], K. C. L. Law [7], L. H. Thompson[7], N. Moriarty[7], C. L. Parish [7], R. J. Williams [8], C. J. Jackson [4,9,10] ✉ & D. R. Nisbet[1,2,3] ✉

Injectable biomimetic hydrogels have great potential for use in regenerative medicine as cellular delivery vectors. However, they can suffer from issues relating to hypoxia, including poor cell survival, differentiation, and functional integration owing to the lack of an established vascular network. Here we engineer a hybrid myoglobin:peptide hydrogel that can concomitantly deliver stem cells and oxygen to the brain to support engraftment until vascularisation can occur naturally. We show that this hybrid hydrogel can modulate cell fate specification within progenitor cell grafts, resulting in a significant increase in neuronal differentiation. We find that the addition of myoglobin to the hydrogel results in more extensive innervation within the host tissue from the grafted cells, which is essential for neuronal replacement strategies to ensure functional synaptic connectivity. This approach could result in greater functional integration of stem cell-derived grafts for the treatment of neural injuries and diseases affecting the central and peripheral nervous systems.

Injectable biomaterials that can deliver, support, and integrate stem cells hold significant promise for tissue regeneration, as they can easily be administered to the site of therapeutic need where they rapidly and effectively fill voids to ensure good tissue contact. Indeed, significant developments have been made to tune the physicochemical properties of such materials towards the parameters required by attachment-based cells. We have previously demonstrated that our self-assembling peptide (SAP) hydrogels consisting of a bio-active epitope derived from laminin (IKVAV) promote neuronal differentiation and integration in vivo[1–3]. Furthermore, the ability of the hydrogel to support neuronal stem cells[4] and astrocytes[5] for up to 7 and 28 days, respectively, in vitro, highlights the superior biocompatibility of the material. However, due to their injectable nature and the general lack of vascularisation at the site of injection, they rely on diffusion of oxygen through the tissue to the site of the graft. This is a significant limitation for metabolically active cells, particularly when they have an acute requirement for support during the important early stages before natural angiogenesis mechanisms have begun[6–9], significantly

[1]Laboratory of Advanced Biomaterials, Research School of Electrical, Energy and Materials Engineering, Australian National University, Canberra, ACT 2601, Australia. [2]The Graeme Clark Institute, The University of Melbourne, Parkville, VIC 3010, Australia. [3]Department of Biomedical Engineering, Faculty of Engineering and Information Technology, The University of Melbourne, Carlton, VIC 3053, Australia. [4]Research School of Chemistry, Australian National University, Canberra, ACT 2601, Australia. [5]School of Chemical Sciences, University of Auckland, Private Bag 92019, Auckland 1142, New Zealand. [6]School of Cellular and Molecular Medicine, University of Bristol, Bristol, Bristol BS8 1TD, UK. [7]The Florey Institute of Neuroscience and Mental Health, The University of Melbourne, Parkville, VIC 3052, Australia. [8]Institute for Mental and Physical Health and Clinical Translation, School of Medicine, Deakin University, Warun Ponds, VIC 3216, Australia. [9]Australian Research Council Centre of Excellence for Innovations in Peptide and Protein Science, Australian National University, Canberra, ACT 2601, Australia. [10]Australian Research Council Centre of Excellence in Synthetic Biology, Australian National University, Canberra, ACT 2601, Australia. ✉e-mail: colin.jackson@anu.edu.au; david.nisbet@unimelb.edu.au

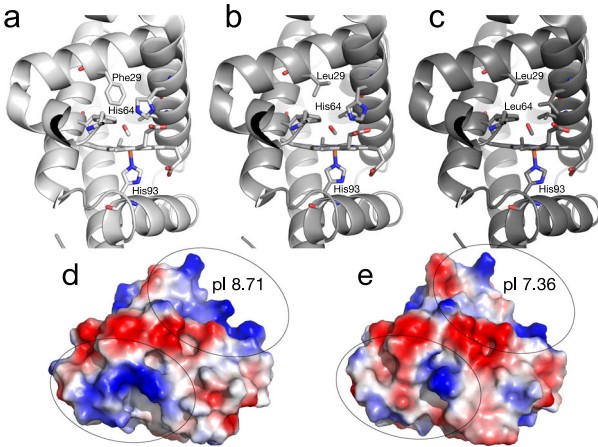

**Fig. 1 | Different myoglobin variants examined in this work. a** The crystal structure of the Leu29Phe mutant of *Physeter macrocephalus* myoglobin (High affinity whale Mb, PDB ID: 2SPL[58]), showing that Phe29 stabilises His64 in a conformation where it can coordinate the bound ligand (in this case CO). **b** The structure of wild-type *Physeter macrocephalus* myoglobin (Sperm whale Mb, PDB ID: 1VXC[59]). **c** The crystal structure of the His64Leu mutant of *Physeter macrocephalus* myoglobin (Low affinity whale Mb, PDB ID: 2MGE[32]), showing that the Leu64 mutation removes the ligand-coordinating imidazole sidechain of His64. The electrostatic surface of *Physeter macrocephalus* myoglobin (Sperm whale Mb) (**d**) and *Equus caballas* myoglobin (Horse Mb)(1AZI[60]) (**e**), showing the greater cationic (blue) surface charge and pI of *Physeter macrocephalus* myoglobin.

limiting any attempt to repair sizeable defects. Indeed, it has been shown that neuronal survival is reduced within a hypoxia or ischemia milieu[10]. However, the technical challenge is significant, as although low oxygen levels (below 2%) can lead to cell death, it is apparent that high levels of oxygen may also induce destructive effects[11,12]. Finally, beyond the need for oxygen for respiration and energy production, oxygen acts as an important signalling molecule for processes that define cell fate, such as apoptosis and differentiation in central nervous system (CNS) cell precursors[13]. It is understood that the major contributor to oxidative stress within the brain is elevated levels of intracellular reactive oxygen species (ROS)[14], which have been shown to promote apoptosis in oligodendrocytes and other CNS derivatives[15]. Altogether, it is clear that homeostasis of oxygen levels within an ideal range is critical for the survival, proliferation, and differentiation of neuronal progenitor cells[12,16].

Oxygen-releasing biomaterials have emerged as a new paradigm for the non-vascular support of cell metabolism within novel engineered tissue in vitro and in vivo[17–19]. While much of the attention has focused on the use of hydrogen peroxide ($H_2O_2$) for controlled and prolonged oxygen delivery[20], it has been recognised that the presence of $H_2O_2$ may result in increased production of free radicals[18], although this can be overcome to some extent via the incorporation of enzymes, such as catalase, in multicomponent systems[19,21]. Fluorinated compounds, specifically perfluorocarbons (PFCs), have also been explored as an oxygen-releasing adjuvant in biomaterials as they are bioinert and can readily dissolve oxygen gas[22]. Despite the potential of PFC functionalised biomaterials, their inability to sustain oxygen levels for the timeframes required prior to vascularisation is a significant drawback[23]. An alternative approach is to utilise naturally occurring oxygen binding proteins, such as myoglobin (Mb), to transport oxygen directly to the tissue. Mb has been used to stimulate oxygenation of stem cells during the engineering of cartilage tissue[17,24]. This solution provides a more promising pathway for a clinically translatable therapeutic, as Mb natively facilitates oxygen transport along partial pressure of oxygen ($PO_2$) gradients to serve as an oxygen reservoir. Mb efficiently binds oxygen via the prosthetic heme group in high oxygen concentrations (forming oxymyoglobin), and releases oxygen in

hypoxic conditions (resulting in deoxymyoglobin), such as those experienced during periods of increased metabolic activity[17,25].

Although proteins, such as Mb, have significant potential in improving cell transplantation outcomes, they cannot be simply injected into the specific site of a stem cell graft to influence cell fate, as they are rapidly degraded and diffuse away from the focal site, and instead required continual infusion[26–28]. Thus, innovations to allow protein delivery and ensure physiologically relevant in vivo presentation and long-term activity are required. Early progress has been made in this area by generating hybrid protein:SAP hydrogels; for example, we recently demonstrated that brain derived neurotrophic factor (BDNF) could be kinetically stabilised[1], increasing its active presentation duration from *ca.* 15 mins to 28 days. Mb has been studied in this context, where it has been imbibed[29] or immobilised within sol/gel films[24,29]. In contrast to covalent protein immobilisation strategies that require chemical modifications, we have recently shown that promotion of non-covalent electrostatic protein:hydrogel interactions under mild conditions can allow for substantial active protein retention within biocompatible SAP hydrogels for long durations[30].

We hypothesised that a hydrogel with a proven stem cell delivery capacity can act as a reservoir to bind and release oxygen via the incorporation of myoglobin and could dramatically improve the efficacy of the stem cell grafting technology within CNS when applied to the site of the graft. In this work, we describe the synthesis of a multifunctional oxygen-responsive hydrogel that is capable of enhancing the delivery and promoting the long-term survival and integration of grafted cortical neural stem cells in the brain. Importantly, these results were achieved within hostile brain tissue that had suffered an iatrogenic injury. Histological analysis at 28 days post in vivo delivery revealed significantly enhanced survival and differentiation of grafted neuronal stem cells towards mature neurons, compared with a Mb-free hydrogel. We observed extensive innervation of the endogenous tissue in the presence of Mb, providing the first evidence of the importance of incorporating bioinspired oxygen delivery within a functional hydrogel to synergistically promote the long-term survival and integration of stem cell transplants. Finally, we show through the use of Mb mutants with different $O_2$ affinity, that $O_2$ release is correlated with functional integration, providing a molecular explanation for our observations. This represents a generalisable strategy for the development of tissue mimetic, readily injectable nanomaterials that have diverse applications, including in cell transplantation, gene and drug delivery, 3D in vitro disease models and organ on a chip technology.

## Results

### Generating a series of myoglobin variants with distinct biochemical and biophysical properties

Mb is an excellent model system to study as an oxygen vector in a hydrogel because it provides design flexibility through its previous characterisation and the availability of several variants with a broad distribution of oxygen affinities. We selected wild-type sperm whale myoglobin (*Physeter macrocephalus*), as well as the Leu29Phe mutant (which increases oxygen affinity by supressing His64 fluctuations away from H-bonding with bound oxygen[31]) and the His64Leu mutant (which has lower affinity for oxygen because the oxygen binding His64 sidechain is replaced with leucine[32]) (Fig. 1a–c). The Leu29Phe mutation increases oxygen affinity 13-fold from wild-type and the His64Leu mutation decreases oxygen affinity 55-fold from wildtype, altogether spanning an oxygen affinity ($P_{50}$ mm Hg) of almost three orders of magnitude (0.007–53; 757-fold)[33,34]. The low affinity variant is likely to release oxygen rapidly in physiological conditions, and the high affinity variant will not release bound oxygen until the environment becomes more hypoxic (low [$O_2$]) and would thus be expected to release $O_2$ more slowly. Wild-type sperm whale myoglobin was mutated using site directed mutagenesis to generate the Leu29Phe and His64Leu mutants, and all three were heterologously expressed in

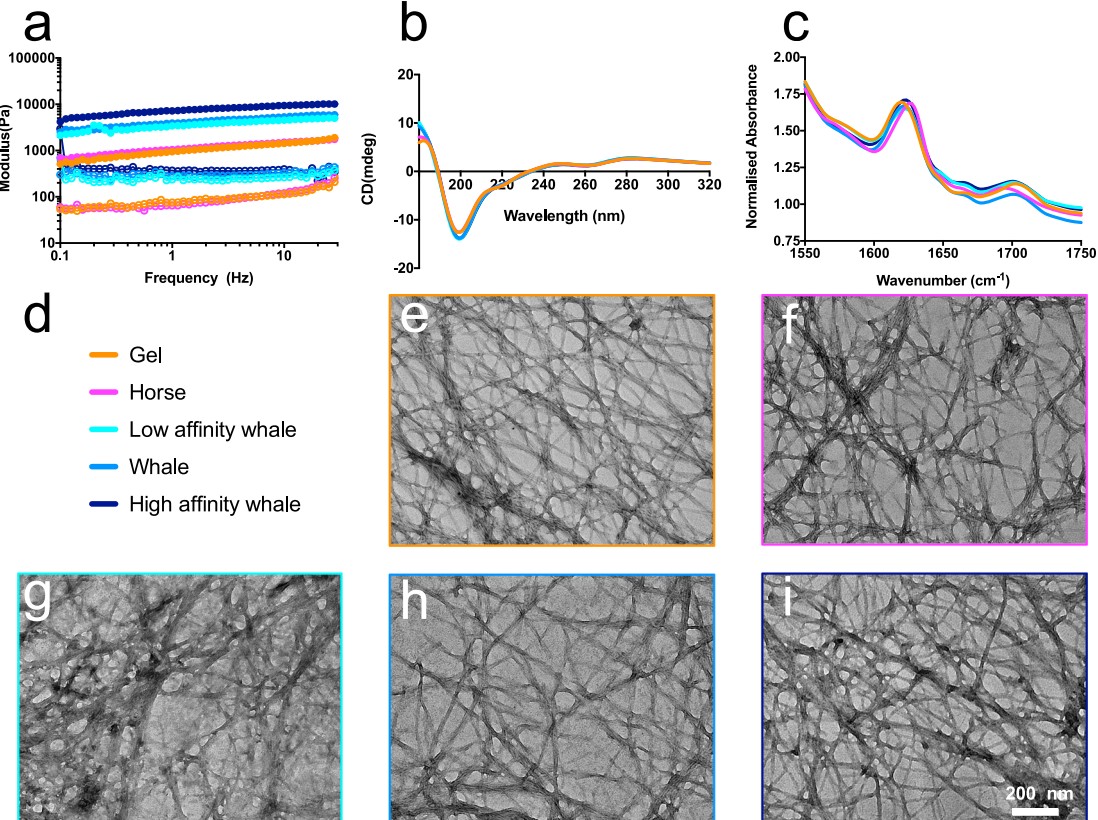

**Fig. 2 | Biophysical and biochemical characterisation of hydrogel materials.**
**a** Rheology data of various Mb:hydrogel preparations showing the storage modulus (solid dots) and loss modulus (empty dots) with a slight change in stiffness.
**b** Circular dichroism absorbance spectra of the hydrogel groups showing the proteins all retain similar secondary structure. **c** FTIR spectra of the hydrogel groups showing the secondary structure containing predominantly β-sheets are formed. **e–i** Representative TEM images of the hydrogel and Mb hydrogels, coloured according to the legend (**d**). Scale bar is indicated in the image. The representative images in (**e–i**) show a summary of at least three independent experiments. Source data are provided as a Source Data file.

*Escherichia coli* and purified to homogeneity (Supplementary Fig. 1). Finally, wild-type horse myoglobin was also selected as a final variant because it has almost identical oxygen affinity to sperm whale myoglobin[35], and very similar sequence identity (88% amino acid identity and 99% similarity; Supplementary Fig. 2), yet the surface charge differs (Fig. 1d and e): Mbs from deep diving animals (such as whales) are thought to have evolved high cationic surface charge for electrostatic repulsion to protect against aggregation owing to high concentration of myoglobin in their muscle tissue[36]. Indeed, sperm whale myoglobin has a significantly higher isoelectric point (pI) than horse (8.71 vs 7.36; Fig. 1d and e).

### Peptide-based nanoscaffolds sustain myoglobin and subsequently oxygen delivery

In order to develop an oxygen vector capable of accelerating stem cell therapies, we optimised a previously described peptide hydrogel, making it capable of delivering oxygen via Mb within an infarct injury. The hydrogel used in this work presents the functional epitope encoding the binding domain of laminin (Fmoc-DD*IKVAV*) in high density on the surface of the nanofibrillar molecular substructure[1,5,37]. This epitope was chosen due to its ability to promote neural adhesion, proliferation, differentiation, and plasticity[4]. We have previously demonstrated that this peptide spontaneously self-assembles into a hydrogel via peptide enabled assembly and amphiphilic re-organisation[38]. We first investigated whether the self-assembly of the hydrogel would be adversely affected by the addition of purified Mb (*Equus caballus*) to the peptide precursor solution immediately before the establishment of the supramolecular substructure responsible for gelation. Using a final mass ratio of 1:15 Mb:peptide, we obtained

characteristically red-coloured hydrogels, which were then optimised to be compliance matched to that of the rodent brain (G″ = 100–1000 Pa[39]) (Supplementary Fig. 3).

The addition of Mb (*E. caballus* and *P. macrocephalus*) resulted in minimal changes in the biophysical properties of the hydrogel substructure as determined by Rheology, circular dichroism (CD), Fourier transform infrared spectroscopy (FTIR) and transmission electron microscopy (TEM) (Fig. 2). These results indicated that the system undergoes a two component assembly, where Mb molecules associate with the surface of, but do not disrupt the structure of, the nanofibrils in a macromolecular association. Indeed, the elasticity, secondary structures, and morphology of the samples are all essentially the same. SAXS analysis of the hydrogel and horse Mb:hydrogel was then undertaken to examine the structure of the hybrid protein:peptide hydrogels in more detail (Supplementary Fig. 4). Both samples exhibited classic hydrogel SAXS scattering profiles, with a slope that is consistent with randomly oriented fibres. Weak features present at −0.16 Å⁻¹ and 0.27 Å⁻¹ in both samples, indicating similar cluster sizes in both of the hydrogels. As these features are present in both the hydrogel and horse Mb:hydrogel samples, this indicates that the base structure of the self- assembled peptide (SAP) hydrogel is unaffected by Mb. There is a slight difference between the two samples, with additional scattering visible between 0.01 Å⁻¹ and 0.15 Å⁻¹ in the horse Mb:hydrogel sample, which is consistent with the scattering of Mb. Overall, this demonstrates that the network morphology underpinning the hydrogel is not affected by the Mb and the data were consistent with a gel in which Mb molecules are distributed as extant structures.

The effect of the gel association on the $O_2$-binding of myoglobin was investigated over 10 h using UV-vis spectroscopy[40]

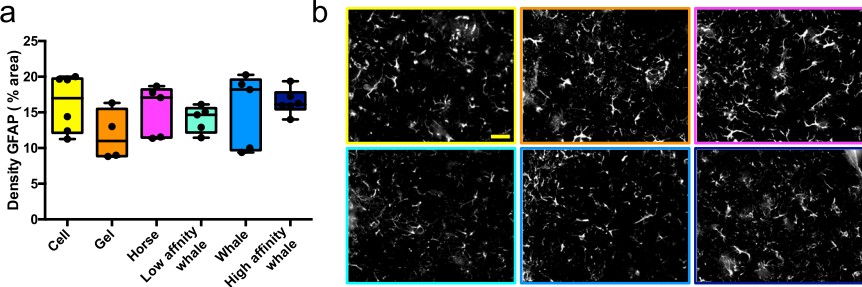

**Fig. 3 | Myoglobin had no effects on the host inflammatory response. a** Density of GFAP + reactive astrocytes surrounding the GFP⁺ graft. Boxes show medians with interquartile ranges, whiskers: min to max show all points. **b** Representative images of GFAP + immunolabeling adjacent to GFP + graft in cell only, hydrogel and Mb:hydrogel groups, respectively. Data are presented as mean ± standard error of the mean (SEM). Scale bar represents 50 μm with all micrographs taken at the same magnification. "Cell" group ($n = 6$) consisted only of transplanted GFP + neural progenitor cells and "Gel" group ($n = 4$) included GFP + neural progenitor cells embedded within Fmoc-DDIKVAV hydrogel. All Mb groups, "Horse" ($n = 5$), "Whale", "Low affinity whale" ($n = 5$) and "High affinity whale" ($n = 6$), included GFP + neural progenitor cells, Fmoc-DDIKVAV hydrogel and the respective Mb variant. Source data are provided as a Source Data file.

(Supplementary Fig. 3). For the Mb to be functional, oxygen must be stabilised within the binding pocket by a ferrous iron ($Fe^{2+}$) covalently bound to a heme prosthetic group. The change in the oxidation state of iron within Mb yields a characteristic absorbance spectrum, which provides an indication that the proteinaceous portion of Mb is stable and in a functional state as it successfully directs oxygen to the heme binding pocket. The hydrogel containing no Mb displayed limited absorption throughout the wavelength range, as expected, while the reduced deoxymyoglobin (horse) spectra at $t = 0$ exhibited a characteristically intense Soret band at 426 nm[41]. The full transition from reduced deoxymyoglobin, to oxymyoglobin, to oxidised myoglobin (metmyoglobin), observable by an intense band at ~409 nm and a weaker Q-band at 628 nm, took place over the course of 10 h. The observation of these spectral changes, which are the same as those observed during the oxidation of normal myoglobin in solution (Supplementary Fig. 5), demonstrate that Mb is incorporated within the hydrogel in a functional state. Interestingly, the rate of oxidation within the hydrogel was substantially slower than what was observed in the absence of the hydrogel, which suggests that the oxygen diffusion coefficient of the hydrogel is lower, consistent with previous studies of oxygen diffusion through hydrogels[42].

**Myoglobin-functionalisation of nanoscaffolds demonstrates in vivo biocompatibility**

We then investigated the capacity of our hydrogels to act as oxygen vectors to support host tissue post administration. To do this, firstly we explored the most important aspect of any synthetic implant, the biocompatibility of the hydrogel oxygen vectors within the host brain, determining how Mb functionalisation and oxygen delivery impacted the host immune response. The resultant peptide/protein concentration of 10 mg mL⁻¹ was administered through ultrafine glass capillaries into the brain. At 28 days post administration, neither the Fmoc-DDIKVAV peptide hydrogel, nor the hydrogels functionalised with Mb, showed any detrimental impact (i.e. no increase in local immune-responsive cells). This was determined through examination of the density of proinflammatory GFAP + astrocytes immediately adjacent to the site of hydrogel administration within the brain, where no increase in the number of reactive astrocytes was observed (Fig. 3). The density of Iba1+ cells was also observed (Supplementary Fig. 6) with no significant difference ($p < 0.05$) between the cell, hydrogel and Mb:hydrogel groups in terms of microglia reaction. Importantly, we have recently shown in the stroke injured brain that an unfunctionalised IKVAV SAP hydrogel has no influence on the immune system with the same density of reactive astrocytes and microglia compared to sham (saline) injected and injury alone controls[1].

**Sustained oxygen delivery from protein-inspired nanoscaffolds concomitantly provides physical and trophic support to transplanted cortical progenitor cells**

Having established that the Mb-functionalised hydrogel is biocompatible, we then investigated the effect of Mb on the survival and differentiation of transplanted progenitor cells. We encapsulated green fluorescent protein reporter (GFP+) neuronal progenitor cells within the peptide hydrogel (±myoglobin) for co-delivery into the brain. The GFP reporter within the cells enables clear identification of transplanted cells and their neuronal processes within the host brain tissue. Using GFP to delineate the graft core, we observed a significantly increased graft volume ($p < 0.05$), when Horse Mb was incorporated within the hydrogel vs the cell control and no-Mb control (Fig. 4a and c). Interestingly, the whale myoglobin mutants displayed an increasing trend in the graft volume with increasing oxygen affinity of the variants (Fig. 4a and b).

Satisfied that our materials provide a significant benefit to support the graft, we then investigated its integration via the ability of transplanted cells to penetrate into the host brain, as this process is a key indicator of brain repair. Graft integration was assessed at 28 days post administration by volumetric analysis of the GFP + fibres within the surrounding brain parenchyma (Fig. 4b). These results showed that the transplanted GFP + progenitors were capable of integrating into the host tissue, with extensive GFP + fibre growth observed surrounding the graft core and innervating the host tissue (Fig. 4b and c). Again, there was a significant increase in innervation volume ($p < 0.05$) in the horse Mb:SAP hydrogel over the SAP hydrogel without Mb, increasing from $0.82 ± 0.05$ mm³ to $1.08 ± 0.09$ mm³. Likewise, the whale Mb variants demonstrated increasing volume of innervation with increasing oxygen affinity. These results also indicate that the high affinity whale Mb matched that of the horse in regard to both graft and innervation volume.

Furthermore, GFP immunohistochemistry-fluorescence (Supplementary Fig. 7) captured of the surrounding host striatum tissue suggest that axonal growth from cells delivered within the peptide hydrogel was not impeded and that the delivery of oxygen via myoglobin within the graft core did not result in GFP + fibres being restricted within the graft core.

While the increased cell survival and graft core volume observed in the Mb:hydrogel data is beneficial, the ability to replace relevant neuronal circuitry/cell types that have been damaged or lost is more important for the treatment of brain injury[43]. To this end, the generation of new, correctly differentiated, neurons within the graft is essential. To investigate the effect of Mb on cell differentiation, we quantified the total number of NeuN+ cells within each graft 28 days post transplantation (Fig. 5a). In light of the varying graft sizes observed (Fig. 4a and c), we expressed the total number of NeuN+

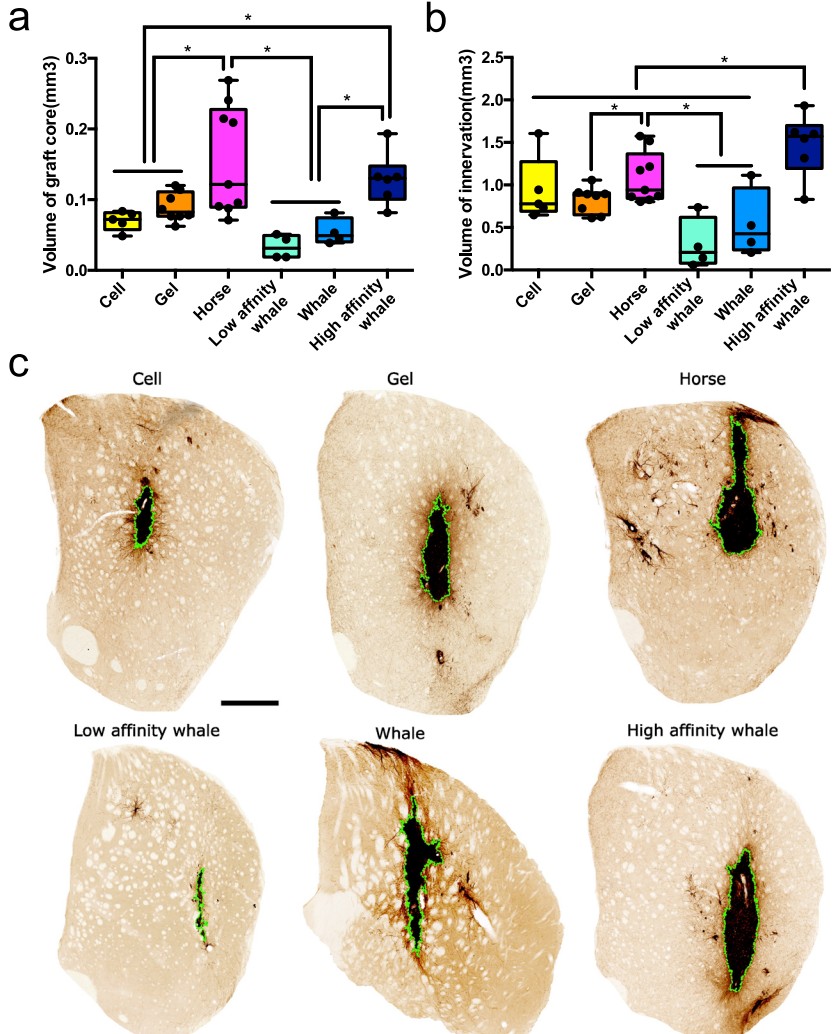

**Fig. 4 | Incorporation of myoglobin within SAPs improves graft survival and graft innervation. a** Volume of graft core in Cell ($n = 5$), Gel ($n = 8$), Horse ($n = 9$), Whale ($n = 4$), Low affinity whale ($n = 4$) and High affinity whale ($n = 6$) groups. Boxes show medians with interquartile ranges, whiskers: min to max show all points. (Cell vs Horse *$p = 0.0195$, Gel vs Horse *$p = 0.0488$, Cell vs High affinity whale *$p = 0.0086$, Gel vs High affinity whale *$p = 0.0246$, Horse vs Low affinity whale *$p = 0.001$, Horse vs Whale *$p = 0.0084$, High affinity whale vs Low affinity whale *$p = 0.025$, High affinity whale vs Whale *$p = 0.0063$) **b** Volume of innervation in Cell ($n = 5$), Gel ($n = 8$), Horse ($n = 9$), Whale ($n = 4$), Low affinity whale ($n = 4$) and High affinity whale ($n = 6$) groups. Boxes show medians with interquartile ranges, whiskers: min to max show all points. (Cell vs High affinity whale *$p = 0.045$, Gel vs

High affinity whale *$p = 0.0069$, Horse vs High affinity whale *$p = 0.0437$, Low affinity whale vs High affinity whale *$p < 0.0001$, Whale vs High affinity whale *$p = 0.0009$, Gel vs Horse *$p = 0.048$, Low affinity whale vs Horse *$p = 0.0029$, Whale vs Horse *$p = 0.0192$) **c** Representative photomicrographs providing a coronal view of GFP + grafts in cell only, hydrogel and Mb:hydrogel groups, respectively. Graft core is delineated in green. Scale bar represents 500 μm. The representative images in 4c show a summary of at least three independent experiments. Data are presented as mean ± standard error of the mean (SEM) (*, $p < 0.05$). One-way ANOVA with Tukey's post hoc test for multiple comparisons and Unpaired two-sided *t*-test were used for statistical analysis. Source data are provided as a Source Data file.

neurons as a proportion of total cells within the graft (i.e. NeuN+GFP +/ GFP + DAPI). We also assessed the percentage of NeuN+ cells compared with undifferentiated GFP + DAPI + cells (Fig. 5b) to provide an indication of the neuronal differentiation and density within the graft. The IKVAV epitope-containing hydrogel resulted in a slight increase in the NeuN number (2801 ± 391 cells/graft) within the graft core compared with the cell only control group (2191 ± 273 cells/graft) (Fig. 5a). This reflects trends we have observed in previously reported work, where we demonstrated that our IKVAV epitope-containing hydrogel scaffolds are capable of increasing the proportion of neurons within the graft[1,2,4]. This is due to the high availability and surface density of the IKVAV epitope in the SAP promotes neuronal adhesion, differentiation, and axonal growth of neural progenitors more efficiently than laminin itself. Remarkably, we observed an additional significant increase ($p < 0.01$) in neuronal cells to a number of 5756 ± 771 cells/mm³ as a

result of the sustained delivery of oxygen via horse Mb (Fig. 5a). In addition, the high affinity whale Mb variant (Leu29Phe) resulted in a significant increase in neuronal differentiation, compared with the controls and lower-affinity Mb variants (Fig. 5b). As the base hydrogel structures are identical (Fig. 2), this suggests that the presence of myoglobin within the hydrogel can not only enhance the delivery and long-term survival of grafted cortical neural stem cells, but also bias their differentiation towards the neuronal fate that is critical for circuity repair and reconstruction. In addition to its biological functionality, this also exemplifies this facile method for the creation of a hydrogel oxygen vector using the immobilisation, of Mb for the controlled release of oxygen in vivo.

To ensure the Mb functionalization did not result in excessive cell proliferation, we further assessed the grafts 28 days post implantation using the expression of Ki67 to mark cells undergoing

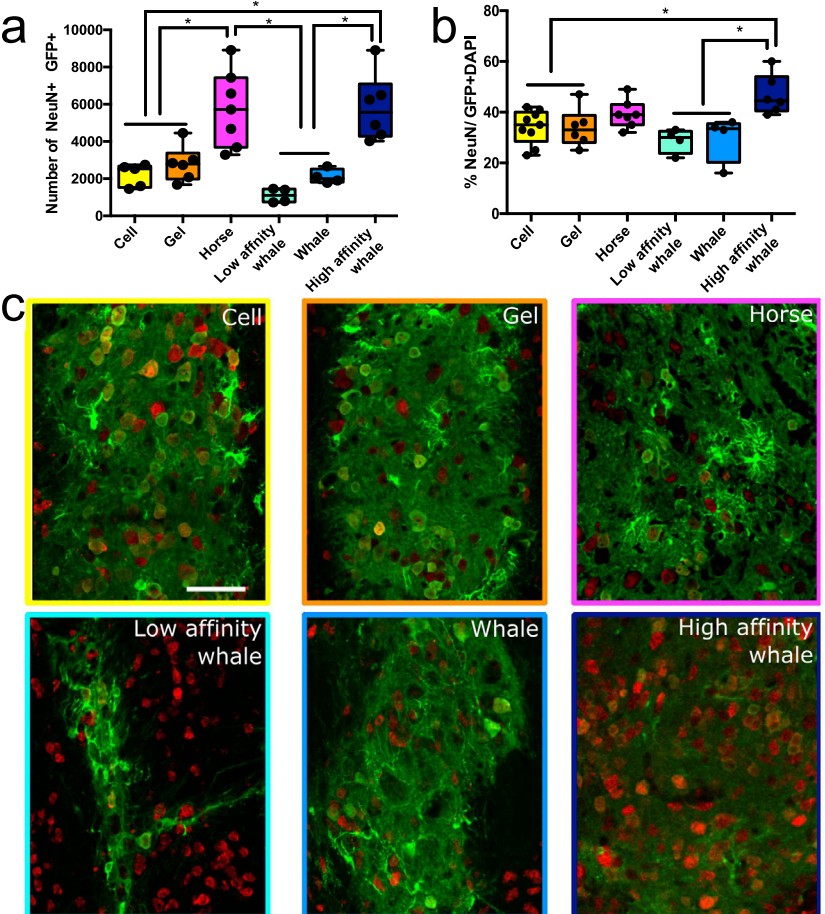

**Fig. 5 | Myoglobin incorporated within SAPs promotes neuronal differentiation. a** Total number of NeuN+ cells in graft in Cell ($n = 5$), Gel ($n = 6$), Horse ($n = 7$), Whale ($n = 4$), Low affinity whale ($n = 4$) and High affinity whale ($n = 6$) groups. (Cell vs Horse *$p = 0.0017$, Gel vs Horse *$p = 0.0072$, Low affinity whale vs High affinity whale *$p = 0.0002$, whale vs High affinity whale *$p = 0.0032$, Horse vs Low affinity whale *$p = 0.001$, Horse vs Whale *$p = 0.0029$, High affinity whale vs Cell *$p = 0.002$, Hight affinity whale vs Gel *$p = 0.0083$) Boxes show medians with interquartile ranges, whiskers: min to max show all points. **b** Percentage of NeuN+ cells in graft compared with undifferentiated GFP + DAPI + cells in Cell ($n = 9$), Gel ($n = 6$), Horse ($n = 7$), Whale ($n = 4$), Low affinity whale ($n = 4$) and High affinity whale ($n = 6$)

groups. (Cell vs High affinity whale *$p = 0.0184$, Gel vs High affinity whale *$p = 0.0323$, Low affinity whale vs High affinity whale *$p = 0.0044$, Whale vs High affinity whale *$p = 0.0079$) Boxes show medians with interquartile ranges, whiskers: min to max show all points. **c** Representative images showing the NeuN + GFP + cells graft in cell only, hydrogel and Mb:hydrogel groups, respectively. Scale bar represents 50 µm. The representative images in (**c**) show a summary of at least three independent experiments. Data are presented as mean ± standard error of the mean (SEM) (*, $p < 0.05$). One-way ANOVA with Tukey's post hoc test for multiple comparison was used for statistical analysis.

proliferation[44]. Excessive proliferation was not observed, with the density of Ki67+ cells and total number of Ki67+ cells being the same between the functionalised and unfunctionalised hydrogels (Supplementary Fig. 8b and c). An increase in doublecortin (DCX+) neuroblasts, which also indicates uncontrolled cell proliferation, was not observed between any of the samples (Supplementary Fig. 8a). Furthermore, although the DCX + migrating neuroblasts were limited to the site of implantation, which is characteristic for fetal tissue-derived neuronal grafts, they are largely concentrated at the periphery of graft. This confirms that the gel does not impede cell migration. Altogether these results suggest that Mb functionalisation of the hydrogel resulted in enhanced integration and differentiation of neural stem cell grafts into existing host parenchyma without excessive proliferation or impedance of cell migration. It is also interesting to note that previous studies have shown that Ki67+ proliferative cells present within human pluripotent stem cell-derived neuronal grafts (matured over 9 months) were predominantly host endothelial progenitors migrating into the graft[1,3]. Further investigation is therefore required to determine whether the Ki67+ cells presented in this study may be endothelial progenitor

cells undergoing angiogenesis within the graft tissue to form new blood vessels.

These results demonstrate the potential of multicomponent nanoscaffolds for advanced regenerative outcomes, which in this case was achieved via the incorporation of oxygen delivery within a functional hydrogel to synergistically promote the long-term survival and, importantly, the integration of stem cell transplants within the host brain. Taken together, these in vivo data suggest the incorporation of Mb in an SAP hydrogel can synergistically promote the long-term survival and, importantly, the integration of stem cell transplants within the host brain, whilst avoiding undesirable cell fates and immune responses.

## Comparison of different myoglobin variants
We next set out to test the parameters governing the beneficial role of oxygen affinity in supporting transplanted stem cell growth and differentiation. As these mutations are within the $O_2$ binding side in the interior of the protein (Fig. 1), they are not expected to affect the properties of the gel, which is confirmed in the biophysical analysis of the gel (Fig. 2). There is no effect on the immune response from any of

the sperm whale myoglobin hydrogels (Fig. 3), nor any increase in cell proliferation (Supplementary Fig. 8). This suggests oxygen release (and the concentrations tested here) does not directly affect these processes. However, when the growth, innervation and differentiation of the transplanted cells are examined, we observe statistically significant differences across this series of low to high affinity Mb. Both the volume of the graft core (3.8-fold) and the level of innervation (4.8-fold) increase significantly from His64Leu, to the Leu29Phe variant (Fig. 4). Likewise, the number of neural cells and level of cell differentiation increases significantly from the low affinity His64Leu variant to the Leu29Phe mutant (Fig. 5). Interestingly, the His64Leu variant performs slightly worse than the gel-only control in terms of graft core volume and volume of innervation ($p < 0.01$), suggesting that premature or early release of the oxygen carried by the myoglobin before oxygen levels have decreased below normal could be deleterious. These results support previous studies that have suggested oxygen homeostasis is essential for neural stem cell survival[11,12], and demonstrates that these Mb:peptide gels can be engineered for improved performance. Importantly, we demonstrate that not only does Mb contribute to improved graft survival owing to its ability to bind and release oxygen in hypoxic conditions, but the levels of oxygen binding and release (engineered homeostasis) can be mediated by the selection of a Mb variants to tune the reservoir of oxygen to the demands of the transplanted cells.

Because the affinity of the Mbs for the gel in vivo will depend on the electrostatic and hydrophobic interactions with the Fmoc-DD*IKVAV* motif, we also sought to test the effect of surface charge (measured as pI) on the performance of the Mbs in the gel. As seen in Figs. 4 and 5, when only the wild-type Mbs are compared, horse Mb exhibits superior performance vs wild-type sperm whale Mb in terms of cell survival and differentiation. This suggests that the higher, cationic, surface charge of sperm whale myoglobin, could result in slightly lower affinity to the hydrogel and leaching from the hydrogel, given the peptide:protein interaction will rely on hydrophobic and potentially anionic interactions with the exposed IKVAV motif.

## Discussion

The future success of cell transplantation technologies will depend on the ability to engineer clinically relevant solutions to improve upon current deployment methods to increase implanted cell survival, while also providing a suitable milieu for the deployed cells to differentiate and integrate within the host circuitry[6,45]. To achieve this, individual facets of the synthetic cellular microenvironments must be engineered to be capable of chemically and physically supporting the survival, differentiation, and neuronal connectivity of transplanted cells[7]. We have recently developed a new class of programmed peptide that fulfils such requirements[2,5,46]. We have investigated their capacity as adjuvants for cell transplantation in brain repair, due to their ability to mimic aspects of brain tissue via the high-density presentation of the functional epitope from laminin (IKVAV), which promotes tissue regeneration and supports cells during all stages of transplantation[3,37]. We have also demonstrated the unprecedented ability of our transplantation vector to protect fragile therapeutics to provide a standardised and supportive milieu unlike the unpredictable injury environment[1,5]. However, this delivery system still faces problems, including poor survival of cells within the core of large grafts, difficulty in promoting lineage specific differentiation, and limited integration of grafted cells within the host tissue. Given the known issues related to oxygen diffusion through hydrogels and the general lack of vascularisation at the site of transplant, the inability to maintain appropriate oxygen homeostasis is one of the most pressing issues.

Here, we provide the first evidence that a tissue-programmed peptide hydrogel containing oxygen carrier proteins, such as Mb and its tuned variants, is capable of sustaining a beneficial supply of oxygen to stem cells grafts in vivo, positively impacting their survival and

integration, and modulating stem cell specification. This was achieved through exploiting the chemically well-defined surface of the thermodynamically stable peptide nanofibrils that underpin our peptide hydrogel (Fig. 2), which acted as a stabilising substrate via the macromolecular self-association of Mb. We have previously demonstrated that this approach protects proteins from proteases, yet maintains their conformation and accessibility to interstitial fluids[1,2]. In terms of the development of strategies to deliver oxygen in situ, it is important to note that the production of the SAP:Mb hydrogel did not compromise the function of either Mb or the hydrogel itself (Fig. 2 and Supplementary Figs. 3, 4 and 5). The increased cell survival, differentiation and integration/innervation that was observed in the Mb:hydrogel cell grafts is compelling evidence that the presence of Mb was beneficial to transplantation (Figs. 4 and 5). It is equally important to note that we did not observe any negative effects relating to the presence of Mb, such as excessive cell proliferation (Supplementary Fig. 8) or immune responses (Fig. 3 and Supplementary Fig. 6). However, the stem cell:hydrogel niche is a highly dynamic environment, with many cells likely to die during implantation whilst others undergo one or more proliferation cycles in situ. It is therefore challenging to quantify the survival rate of those stem cells that were initially transplanted. Further studies to shed light on this could involve, for example, BrdU labelling to track cell proliferation over time, as well as cleaved Caspase-3 to quantify cell death acutely after transplantation.

This work poses two questions relating to the mechanism of how this oxygen-vector hydrogel is contributing to the improved stem cell graft survival, differentiation and integration that we have observed. The first relates to the role of the $O_2$ that is delivered to the cells from the hydrogel once the partial pressure of oxygen drops below a critical value. Wildtype myoglobin serves as an oxygen reservoir within muscle tissue. It will reversibly bind oxygen, extracting the molecule from the blood supply to subsequently release it to the myocytes to maintain an oxygen tension of ~2.5 torr[47]. This is especially important during exercise when the metabolic demand of muscle tissue is high and there is risk of hypoxia. Therefore, we hypothesise that in our application the presence of Mb allows these hydrogels to act as a temporary 'bloodstream' providing acute care for the grafted cells prior to adequate angiogenesis and vascularisation within the graft. The release of oxygen from myoglobin is dependent on the local partial pressure of oxygen within the cell laden material. As such, the cellular respiration will inevitably deplete the myoglobin-facilitated oxygen stores, but ideally by this stage the cells are more mature and angiogenesis/vascular integration has occurred. We have observed host derived angiogenesis over time within large stem cell grafts previously[1,3]. Indeed, the importance of oxygen in enhancing neuronal circuitry reconstruction to ensure functional synaptic connectivity with remote brain regions has been well documented (Fig. 4)[12,16], and this work presents a potential explanation for these observed effects.

Secondly, the delivery of $O_2$ and its function can be defined via the oxygen affinity of the Mb variants, and the strength of the anchoring interactions between them and the scaffolding of the peptide hydrogel. High affinity sperm whale Mb demonstrated better cell survival and differentiation compared to other groups of sperm whale, while low affinity sperm whale Mb showed the poorest cell survival and differentiation (Figs. 4 and 5), indicating the sub-optimal premature release of oxygen has a negative effect on the graft before the oxygen level is under normal level. Finally, the functional differences between wild-type horse and sperm while myoglobin, which have almost identical oxygen affinity and 88% identical amino acid sequences, most likely results from there varying affinity for the hydrogel, resulting in greater loss from diffusion of the more charged sperm while myoglobin. These results shed light on the molecular basis for these results and demonstrate that the system can be easily engineered to optimise for different environments.

In summary, our results demonstrate that the engineered homeostatic environment of our myoglobin laden peptide hydrogel can safely and sustainably maintain the in vivo delivery of oxygen to stem cells grafts, positively impacting cell survival, integration and differentiation. These results demonstrate the importance of engineering hybrid hydrogel systems, capable of concomitantly delivering stem cells with functionally appropriate therapeutics, in this case oxygen, when attempting to induce metabolically demanding tissue repair and reconstruction. Importantly, while proof-of-concept is demonstrated within the brain, this approach represents a rational and generalisable strategy for the development of readily injectable cell supportive nanomaterials for a diverse range of applications, including cell transplantation, gene and drug delivery, 3D in vitro disease models and organ on chip technology.

## Methods

### Institutional review board statement

All animal procedures and methods were conducted in accordance with the Australian National Health and Medical Research Council's published Code of Practice for the Use of Animals in Research and were approved by the Florey Institute of Neuroscience and Mental Health Animal Ethics Committee (protocol code 18-005, February 2018).

**Protein expression, purification, and sequence analysis.** The sequences of *Equus caballus* (Horse Mb, SwissProt accession number: P68082) and *Physeter macrocephalus* (Sperm whale Mb, SwissProt accession number: P02185) were aligned with T-COFFEE server[48] and the pI values were calculated using the Expasy ProtParam Tool[49].

Lyophilised horse myoglobin (Sigma) was reconstituted in phosphate buffered saline (PBS) (pH 7.4) and further purified using size exclusion chromatography (HiLoad 26/600 Superdex 200; Cytiva). The protein eluted as a single dominant peak and SDS-PAGE analysis indicated that the protein was essentially pure (Supplementary Fig. 1). There was one small band at a molecular weight corresponding to dimer, which could have formed during heating of the samples for loading onto the gel; indeed, apo-horse myoglobin is known to form a dimer once the heme is lost[50]. Wild-type *Physeter macrocephalus* myoglobin cloned into pMB413a was a gift from Stephen Sligar (Addgene plasmid # 20058; http://n2t.net/addgene:20058; RRID:Addgene20058[51]). This plasmid was mutated using Gibson assembly mutagenesis[52] to generate the Leu29Phe mutant using the following primers:

F: 5'-GTCGCTGGTCATGGTCAGGACATC*TTC*ATTCGACTGTTCAAATCTCATCCGG-3'

R: 5'-CGGATGAGATTTGAACAGTCG*GAA*TGAAGATGTCCTGACCATGACCAGCGACG-3'

The His64Leu mutant was generated in an indentical manner using the following primers:

F: 5'-GAAAGCTTCTGAAGATCTGAAAAAA*CTG*GGTGTTACCGTGTTAACTGCCCTA-3'

R: 5'-TAGGGCAGTTAACACGGTAACACC*CAG*TTTTTTCAGATCTTCAGAAGCTTTC-3'

Purified DNA from the Gibson assembly reaction was used to transform *E. coli*. The plasmid DNA from a single colony was extracted and the seqeunce of the mutants was confirmed by Sanger sequencing (Ramaciotti Centre for Genomics, Australia).

All proteins were expressed through transformation into BL21(DE3) *E. coli* cells and grown for 20 h at 37 °C with shaking (200 rpm) in 1 L Lysogeny Broth (LB) medium supplemented with 100 mg ampicillin[53]. Cells were harvested through centrifugation ($10,000 \times g$). For lysis, the pellet was resuspended in 25 mM HEPES pH8, 1 mM EDTA, 0.5 mM DTT, lysed by sonication (Sonic Ruptor 400 Ultrasonic Homogenizer (Omni) at 50% power and 50% pulse length for 6 min whilst the cells were immersed in an ice bath) with one repeat after a 6 min recovery at 4 °C. Cell debris was removed by

centrifugation ($30,000 \times g$) for 45 min at 4 °C) and filtration 0.45 μm-pore-size nitrocellulose membrane (Millipore) and the clarified supernatant was collected. Protein was purified through anion exchange chromatography (DEAE fractogel, Merck) equilibrated with 25 mM HEPES pH8, 1 mM EDTA, 0.5 mM DTT, and protein was eluted over a gradient in which the concentration of NaCl in the buffer was increased from 0 to 1.5 M NaCl. The purity of the eluted fractions were then analysed with SDS-PAGE The most pure myoglobin-containing fractions from anion exchange chromatography were then purified further and buffer was exchanged using size exclusion chromatography, which was performed in an identical manner as described for horse myoglobin (above). Pure monomeric fractions from size exclusion chromatography was concentrated to ~16 mg/mL using an Amicon Ultra-15 centrifugal filter 10 kDa (Millipore) and loaded into the column with a syringe. Protein concentration was determined using absorbance at 280 nm using an extinction coefficient of 15470 $M^{-1}$ $cm^{-1}$ for *Physeter. macrocephalus* myoglobin (UniProt acc. number P02185) and 13980 $M^{-1}$ $cm^{-1}$ for *Equus caballus* myoglobin (UniProt acc. number P68082). Representative gels and size exclusion chromatograms are provided in Supplementary Fig. 1. The two-step purification of proteins via affinity and size exclusion chromatography, as performed here, has been shown to be sufficient to remove endotoxin from protein preparations[54].

**Preparation of self-assembling peptide scaffolds.** Fmoc-DDIKVAV was synthesised at 0.4 mmol scale by solid phase peptide synthesis using a rotating glass reactor vessel. All chemicals were purchased from Sigma Aldrich (Australia) with the amino acids being purchased from Pepmic (China). The Fmoc-DDIKVAV hydrogel were prepared at a final concentration of 15 mg mL$^{-1}$ using a well-established pH switch. Briefly, ~10 mg of peptide was dissolved in 200 μL of deionised water with 100 μL 0.5 M sodium hydroxide (NaOH). Then 0.1 M of hydrochloric acid (HCl) was added dropwise with continuous vortexing until the solution reached physiological relevant pH (Oaktron pH 700 micro pH electrode, Thermo Scientific). 0.01 M Phosphate buffered saline (PBS)[55] was added to final 15 mg mL$^{-1}$ concentration of hydrogel. For in vitro use, Hank's buffered saline solution (HBSS) (Gibco) was used in place of the PBS.

For the preparation of the Mb:hydrogel hybrid, the various Mb (in PBS) were added into the hydrogel (prepared as described above) to a final concentration of 1 mg mL$^{-1}$ myoglobin. The vial was vortexed (30 s) for homogenisation and rested (60 s) for re-gelation. The final hydrogel contained 15 mg mL$^{-1}$ peptide hydrogel and 1 mg mL$^{-1}$ myoglobin.

**Spectral characterisation of myoglobin-functionalised hydrogels.** Unfunctionalised Fmoc-DDIKVAV hydrogel (15 mg mL$^{-1}$) and the same hydrogel functionalised (as above) with a final concentration of 1 mg mL$^{-1}$ myoglobin that was already reduced by reaction with excess sodium dithionite ($Na_2S_2O_4$; 17 mM) were prepared in a $N_2$-containing anaerobic hood. Upon removal from the anaerobic hood, UV-Vis absorbance of the samples was monitored using a 96-well plate reading Epoch spectrophotometer (Biotek) over 10 h over a range of 350–700 nm. The same experiment was performed with free myoglobin in solution, in which everything in the system was identical, except for the absence of the hydrogel.

**Transmission electron microscopy (TEM).** Negative-staining transmission electron microscopy (TEM) was performed using a HITACHI HA7100 TEM with a LaB6 cathode at 125 kV (tungsten filament). Formvar-coated copper grids were prepared with electron glow discharge at 15 mA for 30 s. The formvar-coated side of grids was loaded with hydrogel for 30 s, washed with DI $H_2O$ (20 μL), treated with urea-formaldehyde (UF, 20 μL), and finally immersed into UF drop for 30 s. Between each step, excess solution was blotted off using filter paper.

Then, the grids were allowed to dry overnight before imaging in the TEM.

**Fourier transform infrared spectroscopy (FTIR).** Fourier transform infrared spectroscopy (FTIR) was performed using an Alpha Platinum Attenuated Total Reflectance (ATR) FTIR (Bruker Optics) to monitor interactions in Amide I region ($1550-1750$ cm$^{-1}$). Approximately 20 mL of peptide hydrogel was placed on the single reflection diamond. Absorbance scans were obtained for each peptide, and a background buffer scan subtracted.

**Circular dichroism (CD).** Circular dichroism (CD) was performed using a Chirascan CD Spectrometer (Applied Photphisics Limited, Version 4.5.1848.0) to determine the secondary structure of hydrogel. The hydrogel was diluted at 1:10 ratio of hydrogel and DI H$_2$O to reduce scattering effects. The diluted gel around 400 μL was added into the cuvette with a 10 mm pathlength. CD scans ranged from 180 nm to 320 nm with a step size of 0.5 bandwidths using a Chirascan CD spectrometer (Applied Photophysics Limited) and a baseline (DI H$_2$O) was subtracted. The resulting data were averaged and smoothed post-acquisition using Chirascan software(version4.5.1848.0).

**Small angle X-ray scattering (SAXS).** SAXS was performed using SAXS/WAXS beamline at the Australian Synchrotron[56]. Measurements were taken using camera length 900 mm, time exposure 1 second, energy 12 keV and 5% flux. Samples (SAPs groups, SAPs+ Myoglobin groups) were prepared as detailed above 1 day before measurement and stored in Eppendorf vials. PBS buffer was loaded into a 1 mm glass capillary for background measurements. Each hydrogel sample was loaded into six of the same capillaries for measurement. Capillaries were loaded into a custom mount which can hold and move the capillaries in two dimensions, with Kapton film windows. 1 s exposures were taken for each hydrogel-loaded capillary at 10 different positions evenly spread along the 3 mm capillary length. The average of these positions was used for one capillary. Each group of 6 capillaries for each sample was further averaged and the PBS background subtracted using ScatterBrain. There was strong scattering from the Kapton windows centred at ~0.4 Å$^{-1}$, which marks the upper limit of the useable Q-range measured.

**Rheology.** The rheological analysis was performed using a Kinexus Pro+ Rheometer (Malvern) and rSpace software(version 1.72). Approximately 0.2 mL of hydrogel was placed on a 20 mm roughened plate (with solvent trap, Lower Geometry: PLS55 C0177 SS, Upper Geometry: PU20 SR1351 SS). The gap size was 0.2 mm, and multiple frequency sweeps were performed for frequencies ranging from 0.1 to 100 Hz with a 0.1% oscillatory strain at a constant required temperature (37 °C). Each gel was allowed a minimum of 5 min to set before testing.

**In vivo transplantation of cells with myoglobin.** Animals were group housed in individually ventilated cages with *ad libtum* access to food and water. Cells for transplantation were obtained from time mated mice expressing green fluorescent protein (GFP) under the β-actin promoter, which enable clear distinction of the grafted cells (GFP + ) cells within the host brain. Cortical brain tissue was isolated from pups at embryonic day 14.5 (E14.5) and dissociated. Following dissociation of the cortical tissue, cell viability was confirmed to be >90% by trypan blue staining prior to resuspending the cells at a final density of 100,000 cells ul$^{-1}$. Hydrogels were prepared as previously stated and sterilised by UV lamp for 2 h. Stock myoglobin was filtered by syringe filters (0.2 μm). Each gel underwent mechanical sheering via vortexing and was mixed at a 1:1 ratio with cells to a final concentration of 10 mg mL$^{-1}$ of gel and 50,000 cells ul$^{-1}$, respectively, immediately prior to in vivo delivery[57].

Adult C57BL/6 mice ($n = 6$) were anaesthetised with 2% isoflurane and placed in the stereotaxic frame. A craniotomy was performed and unilateral microinjections of cells (total of 100,000 cells) and hydrogels (total of 2 μl) were implanted into the host striatum (0.5 mm anterior and 2 mm lateral to Bregma, and 3 mm below the surface of the brain). After 28 days, mice were killed by an overdose of sodium pentobarbitone (100 mg/kg) and transcardially perfused with warm saline followed by 4 % paraformaldehyde (PFA). Brains were removed, post-fixed for 2 h in 4% PFA and cryo-preserved overnight in 30% sucrose solution. Brains were sectioned on the coronal plane using a freezing microtome (40 μm thickness, 1:10 series).

## Immunohistochemistry & quantification

Immunohistochemistry was performed on free-floating brain sections as previously described[3]. In brief, brain sections were washed and incubated in primary antibodies overnight at room temperature, including: rabbit anti-GFP (1: 20,000; Abcam, ab290), chicken anti-GFP (1: 1,000; Abcam,ab13970), sheep anti-Ki67 (1:40, R&D Systems, AF7649), goat anti-doublecortin(DCX) (1:1000, Santa Cruz, sc-8066), chicken anti-GFAP (anti-glial fibrillary acidic protein, 1:500, Novus biological, NBP1-056198), mouse anti-NeuN (1:1000;Abcam, ab104224), rabbit anti-iba1 (1:1000, WAKO, 019-19741). The following day sections were rinsed and blocked in 5% donkey serum for 20 min. Secondary antibodies for (i) direct detection were used at a dilution of 1:200−DyLight 488, 549 or 649 conjugated donkey anti-mouse, anti-sheep, anti-chicken or anti-rabbit (Jackson ImmunoResearch); and (ii) indirect with streptavidin-biotin amplification−biotin conjugated donkey anti-rabbit (1:500; Jackson ImmunoResearch, 711-065-152) followed by peroxidase conjugated streptavidin (Vectastain ABC kit, Vector laboratories). Finally, fluorescently labelled sectioned were stained with 4′, 6-diamidino-2-phenylindole (DAPI, 1:5000, Sigma-Aldrich) to enable visualisation of all cells. Sections were mounted onto gelatinised slides and coverslipped. All fluorescent images were captured using a Zeiss Axio Observer.Z1 epifluorescence and bright images were obtained using a Leica DM6000 upright microscope.

Total number of NeuN, DAPI, Ki67 and DCX cells within the graft, and the percentage of NeuN+, as well as density of Ki67+ cells in grafts were counted from images captured at ×40 magnification and expressed as per mm$^3$. Host-derived GFAP$^+$ density (assessed as % immunoreactive pixels) were assessed at the graft-host border, as previously described and analysed[1].

Volumetric assessments of the graft were performed as previously described[35]. In brief, GFP-labeling was used to delineate the graft from the host tissue. Tissue was serial sectioned, and volume was calculated according to Cavalieri's principal using section thickness, sum of areas, and interval. The volume of graft core is identifiable by the cell body and any innervation of fibres are excluded. Volume of innervation is assessed by excluding the graft core from the analysis.

## Statistical analysis

Data shown are means ± standard error of the mean (SEM). Data were analysed using Graph Pad Prism 6.0 by one-way ANOVA with Turkey post hoc statistic testing for multiple comparisons or by a two-tailed t-test. For analysis of Iba1+ labelled cells, a one-way ANOVA with post hoc Bonferroni was used. Differences at $p < 0.05-0.01$ were considered statistically significant.

## Reporting summary

Further information on research design is available in the Nature Portfolio Reporting Summary linked to this article.

## Data availability

The atomic coordinates and structure factors for the Leu29Phe mutant of *Physeter macrocephalus* myoglobin (High affinity whale Mb, PDB ID: 2SPL), wild-type *Physeter macrocephalus* myoglobin (Sperm whale Mb,

PDB ID: 1VXC), the His64Leu mutant of *Physeter macrocephalus* myoglobin (Low affinity whale Mb, PDB ID: 2MGE) are available from the Worldwide Protein Data Bank (http://wwpdb.org/). All data during and/or analysed during the current study are available within the paper and Supplementary Files, or available from the corresponding authors on reasonable request. The source data underlying all Figures and Supplementary Figures are provided as a Source Data file. Source data are provided with this paper.

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

## Acknowledgements

We would like to acknowledge the generous gift from Mr. Tom Barbieri and Mrs. Vivienne Barbieri that has contributed towards this study. This is greatly appreciated. Access to the facilities of the Centre for Advanced Microscopy (CAM) with funding through the Australian Microscopy and Microanalysis Research Facility (AMMRF) is gratefully acknowledged. This work was funded by an NHMRC Ideas Grant GNT 1185094 to D.R.N. and C.J.J. and an Australian Research Council Discovery Grant DP220102549 to D.R.N. and C.J.J.

## Author contributions

D.R.N. and C.J.J. contributed equally to this work. D.R.N., R.J.W. and C.J.J. provided conceptualisation, data analysis, supervision. Y.W. and E.R.Z. performed the experiments and all data analysis. N. H., A.P. and L.L.T. performed protein expression, purification, and sequence analysis. C.L.P., K.C.L.L., L.H.T. and N.M. implanted animals. D.R.N., C.J.J., C.L.P., R.J.W., Y.W. and E.R.Z. discussed the results, wrote, reviewed, and edited the paper. All authors have read and agreed to the published version of the paper.

## Competing interests

The authors declare no competing interests.
