## [Peer Review File · Nature Communications]

Reviewer comments, first round review -

Reviewer #1 (Remarks to the Author):

This is an interesting study which focusses on developing a myoglobin laden peptide hydrogel with the aim of sustaining the in vivo delivery of oxygen to stem cells grafts, positively impacting cell survival, integration and differentiation. The study presents some novelties- particularly on the use of myoglobin for this application. It also builds on an existing platform developed by the applicants. Nonetheless it is not clear that the paper has sufficient novelty to achieve the impact required by this journal nor that the results are as compelling as the authors claim. The manuscript is also quite confusing in places with insufficient rationale provided for the different myoglobins used or cell types delivered. In addition, the applicants have not convincingly demonstrated that the changes seen, which are more modest than the applicants claim, are down to the increased oxygen carrying capacity of the platform.

Reviewer #2 (Remarks to the Author):

In this manuscript, the authors demonstrated a novel method to fabricate injectable, oxygen-releasing biomaterials that can be used as carriers for cell-based therapies and evaluate the utility of this biomaterial in a rodent model of stroke. This work is highly innovative, the experiment methods sound, and the results potentially beneficial for neural tissue regeneration. However, there are a few points in the manuscript that warrant clarification, detailed below.

1. There have been several reports of fluorinated biomaterials as oxygen-releasing biomaterials for tissue engineering in the past. While authors discuss using peroxides to create oxygen-releasing biomaterials and discuss why their strategy may have advantages, it would be useful to also discuss how the materials reported in the manuscript compared to fluorination strategies.
2. It is confusing that only horse Mb is discussed before referring to Figures 1-4, as whale Mb is shown in all these figures. It is not clear from the text that horse and whale Mb are being compared until after Figure 5 has been discussed. I suggest reorganizing the results section to cover an entire figure at once rather than jumping around.
3. Please add scale bars to images in Figure 2. It is also not clear where the grafts and stroke cavities are. Please clarify. Finally, while GFAP reactivity appears equivalent across the conditions shown, is this level of activation similar to the injury alone or does providing cells and/or material vary from injury alone?
4. Figure 3, how was the volumes quantified? How were graft cores defined? The photomicrographs make it quite difficult to tell what graft core or innervation looks like. It would be helpful to draw lines around where the graft-tissue boundaries are.
5. Please add scale bars to images in Figure 3.
6. In Figure 4, it is not clear which graft condition is being shown in the images in panels C and D.
7. Figure 4D, it seems the cell alone graft has more double positive cells than all other groups.
8. When discussing Supplementary Figure S4a, the authors state that cells do not migrate from the graft. Is this desirable for tissue integration? If so, why? Please expand on this interpretation in the text.
9. NeuN+/GFP+ cells are quantified to show transplanted cells differentiated to neurons. However, can the authors discuss how well transplanted neural stem cells survive?
10. Any effects on immune cell response? Initial events that trigger regeneration?
11. Any potential concerns of immune rejection when using xeno-derived myoglobin?
12. Are there any observations or thoughts about how oxygen delivery affects graft/material angiogenesis/vascular integration over time?
13. How long are the biomaterials expected to be capable of oxygen release in vivo?

Reviewer #3 (Remarks to the Author):

The manuscript by Wang et al. is aiming to solve the lack of oxygen within cell grafts after transplantation into the brain and thus increase cell survival. The proposed solution of using an engineered hybrid hydrogel, containing oxygen-binding myoglobin, is innovative and interesting, and the results seem promising. The platform has the potential to be used for a variety of injuries and neurodegenerative conditions. We recommend major revisions to address the following issues:

1. The rationale behind the groups "horse", "low affinity whale" etc. is only explained during the last section of the results, and is quite confusing throughout the paper. Please either move that section earlier in the results, or explain in the text the differences between the groups and the figures legends.

2. In the results section, the authors mention: "... compliance matched to that of the rodent brain ($G' \approx 100$ Pa, $G'' = 100$ Pa)". Please provide a reference or results to support the mentioned values.

3. Please better explain the results in supplementary figure 3 and add a legend. What indicates the functional state of the Mb, with and without the gel?

4. In the biocompatibility section:

a. To support the statement of "no influence on the immune system", please describe the microglial reaction to the hydrogel.

b. Please explain the experimental groups better: Did the "cell" group have no gel? Did other groups all contain cells and gel? Please address this in the text as well.

c. Figure 2, legend, line #192 – where is image C? Please correct.

5. "Sustained oxygen delivery" section:

a. The cells are called "cortical progenitor cells" (line # 198), yet they are being transplanted into the striatum. Please explain the rationale.

b. To strengthen the results, it would be great to add high magnification images of the GFP+ fibers that integrated into the host tissue (lines # 215-217).

c. Lines 239-241 – the authors state that "Cell control and the IKVAV epitope-containing hydrogel resulted in no difference in NeuN number within the graft core". It seems that this result contradicts earlier results by the group. Please add a possible explanation to the manuscript.

d. A better explanation on the difference between Fig. 4A and Fig. 4B would benefit the paper.

e. Figure 4D – In some images (for example yellow) most cells co-localize (GFP+ and NeuN+), whereas in others (the whale groups), most NeuN+ seems to be GFP-. Can you determine whether those cells are transplanted cells, and not invading host cells?

f. Line 274 – Did the authors stain the tissue for endothelial markers, to determine which cells express the Ki67? Otherwise, please rephrase.

6. When transplanting cells in the gels:

a. Were the cells characterized after isolation/before transplantation? Was cell viability assayed before mixing with the gel? Please add those results.

b. Do injections of 2ul into the healthy brain tissue cause damage?

c. Did the authors test for endotoxins in the gels and the Mb (produced in E.Coli)?

d. Can the cells survive in the gels in vitro?

7. Last section of the results should be earlier in the paper to explain the rationale and the experiments.

a. When sacrificing the animals at 28 days, was the gel still present in the tissue?

b. For how long can Mb release oxygen?

8. In the discussion, lines # 394-397 – Did the authors look for neo-vascularization? Would be interesting to see.

9. In the methods section:

a. Line # 436 – please add figure number.

b. In vivo transplantation:

i. Were the gels prepared before mixing with the cells?

ii. What was the final concentration of the gels?

iii. How many cells were transplanted per animal?

We thank all reviewers for their valuable comments concerning our manuscript entitled
"Engineered homeostasis: hydrogel oxygen reservoirs increase functional integration of
neural stem cell grafts by meeting their metabolic demands". We have taken all criticisms
of the paper onboard and have revised accordingly. Particularly we have now provided
additional rationale for our selection of myoglobin variants. We have also rearranged the
results (as suggested) to help improve the clarity of the manuscript. The response to the
reviewer comments are as follows:

**Reviewer #1 (Remarks to the Author):**

This is an interesting study which focusses on developing a myoglobin laden peptide
hydrogel with the aim of sustaining the in vivo delivery of oxygen to stem cells grafts,
positively impacting cell survival, integration and differentiation. The study presents
some novelties- particularly on the use of myoglobin for this application. It also builds
on an existing platform developed by the applicants. Nonetheless it is not clear that the
paper has sufficient novelty to achieve the impact required by this journal nor that the
results are as compelling as the authors claim. The manuscript is also quite confusing in
places with insufficient rationale provided for the different myoglobins used or cell types
delivered. In addition, the applicants have not convincingly demonstrated that the
changes seen, which are more modest than the applicants claim, are down to the
increased oxygen carrying capacity of the platform.

This reviewer comment is somewhat vague and therefore difficult to directly
address. We have interpreted the need for improved organisation of the
manuscript (also highlighted, and with greater clarity and detail, by reviewers 2 and
3). This improved organisation of the results and manuscript at large, is detailed
below in response to both of these reviewers.

Regarding the reviewers comment of lack of evidence to the changes observed
being a consequence of increased oxygen, the entire paper is focused on this (and
was not an opinion shared by either of the other reviewers, who provided specific
and very useful comments that allowed us to improve the paper.

**Reviewer #2 (Remarks to the Author):**

In this manuscript, the authors demonstrated a novel method to fabricate injectable,
oxygen-releasing biomaterials that can be used as carriers for cell-based therapies and
evaluate the utility of this biomaterial in a rodent model of stroke. This work is highly
innovative, the experiment methods sound, and the results potentially beneficial for
neural tissue regeneration. However, there are a few points in the manuscript that
warrant clarification, detailed below.

1. There have been several reports of fluorinated biomaterials as oxygen-releasing
biomaterials for tissue engineering in the past. While authors discuss using peroxides to
create oxygen-releasing biomaterials and discuss why their strategy may have
advantages, it would be useful to also discuss how the materials reported in the
manuscript compared to fluorination strategies.

Thank you for this suggestion. We have incorporated the following into the
introduction (Ln 75-79):

"...Fluorinated compounds, specifically perfluorocarbons (PFCs) have also been
explored as an oxygen-releasing adjuvant in biomaterials as they are bioinert and
can readily dissolve oxygen gas(Chin et al., 2008). Despite the potential of PFC
functionalised biomaterials, their inability to sustain oxygen levels for the
timeframes required prior to vascularisation is a significant drawback (Farris et al.,
2016)..."

2. It is confusing that only horse Mb is discussed before referring to Figures 1-4, as whale
18 Mb is shown in all these figures. It is not clear from the text that horse and whale Mb are
19 being compared until after Figure 5 has been discussed. I suggest reorganizing the
20 results section to cover an entire figure at once rather than jumping around.

Thank you for this suggestion. We have significantly restructured the manuscript to
introduce all systems at the start and explain how they differ as requested. This
constituted a significant re-write throughout the manuscript that is too large to
paste here, but we have uploaded a track change document to aide the reviewer in
identifying this revision. We believe this has significantly improved our manuscript.

3. Please add scale bars to images in Figure 2. It is also not clear where the grafts and
stroke cavities are. Please clarify. Finally, while GFAP reactivity appears equivalent across
the conditions shown, is this level of activation similar to the injury alone or does
providing cells and/or material vary from injury alone?

We have updated Figure 2 (now Figure 3) as requested and included below.

We have also added the following "...Scale bar represents 50 μm with all micrographs
taken at the same magnification..."

Regarding the GFAP activation query, the following has been included in the
manuscript:

"Importantly, we have recently shown that an unfunctionalized IKVAV SAP hydrogel
has no influence on the host immune system with the same density of reactive
astrocytes and microglia as sham (saline) injected animals (Nisbet et al., 2018)."

4. Figure 3, how was the volumes quantified? How were graft cores defined? The
photomicrographs make it quite difficult to tell what graft core or innervation looks like.
It would be helpful to draw lines around where the graft-tissue boundaries are.

Thankyou. We have updated Figure 3 (now Figure 4) as requested.

We have updated the text in the methods section to clarify this aspect of the
analysis (Ln 639-644):

"Volumetric assessments of the graft were performed as previously described (Bye et
al., 2012)³⁶. In brief, GFP-labelling was used to delineate the graft from the host
tissue. Tissue was serial sectioned, and volume was calculated according to
Cavalieri's principal using section thickness, sum of areas, and interval (Cavalieri
1966). The volume of graft core is identifiable by delineation around the
immunolabeled cell bodies, excluding the emanating fibres. Volume of innervation
was assessed by delineation of the graft inclusive of all immunolabeled fibers
emanating from the graft core."

5. Please add scale bars to images in Figure 3.

Thankyou. We have updated the figure as requested.

6. In Figure 4, it is not clear which graft condition is being shown in the images in panels
C and D.

Thank you for bringing this to our attention. Upon review of Figure 4 (now Figure
5), we believe that panel C is unnecessary as panel D clearly demonstrates the
proportion of differentiated neurons. We have updated the Figure 4 (now Figure 5)
and the associated description accordingly. We have also selected more
representative micrographs. The changes in the results discussion are in several
different places making it difficult to detail them. However, we have uploaded a
track change version of the document where they can be seen. This was a great
point, thank you. We have included the revised caption below.

Figure 5: Myoglobin incorporated within SAPs promotes neuronal differentiation. A: Total number of NeuN+ cells in graft. B: Percentage of NeuN+ cells in graft compared with undifferentiated GFP+DAPI+ cells. C: Representative images showing the NeuN + GFP+ cells graft in SAPs and SAPs+Mb groups with different oxygen variants, respectively. Scale bar represents 50 μ m. Data represents mean \pm standard error of the mean (SEM).

7. Figure 4D, it seems the cell alone graft has more double positive cells than all other
groups.

This was a good pick up by the reviewer and we apologise for not including the most representative image for the panel. The figure (now Figure 5) has been updated with a new, replacement image of a graft in the presence of the High affinity whale myoglobin hydrogel.

8. When discussing Supplementary Figure S4a, the authors state that cells do not migrate from the graft. Is this desirable for tissue integration? If so, why? Please expand on this interpretation in the text.

We agree with the reviewer's feedback that the discussion of Supplementary Figure S4a does not clearly reflect our intent for this figure. In essence, we aimed to demonstrate that the gel does not cause the grafted cells to proliferate out of control, as indicated by an increase in KI67 proliferative cells. This is also observable by an increase in DCX+ neuroblasts.

We also set about to confirm that the gel did not impede cell migration. Cell
migration within fetal tissue-derived neuronal grafts is notably limited to the site of
implantation. Nonetheless, the DCX+ migrating neuroblasts can be seen
concentrating around the edge of the graft as seen in our previous work (Soma et
al., 2017), indicating that cell migration is not impeded by the gel. However, most
evidence of a graft's ability to integrate is the survival of the grafted neural
progenitors, their maturation and critically, their ability to extend axonal processes
into and synapse with the host tissue. The images depicted in Figure 3 (now Figure
4) importantly highlight graft survival and evident GFP+ neurite extension into the
host parenchyma.

This has been updated in the text:

"...To ensure the Mb functionalization did not result in excessive cell proliferation,
we further assessed the grafts 28 days post implantation using the expression of
Ki67 to mark cells undergoing proliferation⁴⁸. Excessive proliferation was not
observed, with the density of Ki67+ cells and total number of Ki67+ cells being the
same between the functionalised and unfunctionalized hydrogels (**Supplementary**
**Figure S8b and S8c**). An increase in doublecortin (DCX+) neuroblasts, which also
indicates uncontrolled cell proliferation, was not observed between any of the
samples (**Supplementary Figure S8a**). Furthermore, although the DCX+ migrating
neuroblasts were limited to the site of implantation, which is characteristic for fetal
tissue-derived neuronal grafts, they are largely concentrated at the periphery of
graft..."

9. NeuN+/GFP+ cells are quantified to show transplanted cells differentiated to neurons.
However, can the authors discuss how well transplanted neural stem cells survive?

This is a challenging question for the field and near impossible to quantify. While we
know how many cells were transplanted (50,000 cells/ul), many cells die during
implantation, and added to this the implanted neural progenitors can undergo 1 or
several rounds of proliferation in situ. Most important is the relative comparison
across groups.

We have incorporated the following into our discussion:

"...However, the stem cell:hydrogel niche is a highly dynamic environment, with
many cells likely to die during implantation whilst others undergo one or more
proliferation cycles in situ. It is therefore challenging to quantify the survival rate of
those stem cells that were initially transplanted. Further studies to shed light on
this could involve, for example, BrdU labelling to track cell proliferation over time,
as well as cleaved Caspase-3 to quantify cell death acutely after transplantation...."

10. Any effects on immune cell response? Initial events that trigger regeneration?

We have now included within our supplementary information the microglia (Iba1+) cell density (Supplementary Figure S6) to compliment the findings from our astrocyte (GFAP+) cell density analysis. No significant difference was found between the groups, indicating that the presence of the hydrogel and Mb:hydrogel groups did not elicit an increased immune response that may impact neuronal survival and maturation.

This has been added to the text:

“...The density of Iba1+ cells was also observed (**Supplementary Figure S6**) with no significant difference ($p < 0.05$) between the cell, hydrogel and Mb:hydrogel groups in terms of microglia reaction....”

We have also updated our methods section to include Iba1+ analysis.

Iba1 Rabbit WAKO 019-19741 1:1000

11. Any potential concerns of immune rejection when using xeno-derived myoglobin?

While the myoglobin was expressed heterologously in *E. coli*, we purified the myoglobin using affinity chromatography and size exclusion chromatography, which has been demonstrated to separate endotoxin from proteins (Petsch, 2000). Consistent with this, we also see no indication of immune response due to the presence of endotoxin (**Figure 3**).

This has been added to the text:

"...The two-step purification of proteins via affinity and size exclusion
chromatography, as performed here, has been shown to be sufficient to remove
endotoxin from protein preparations..."

In terms of myoglobin itself, it is highly similar to mouse myoglobin, and we do not
expect to see a significant immune response, as is consistent with the reactive
astrocyte and microglia response (Figure 3 and Supplementary Figure S6).
Additionally, the Mbs are impregnated within the hydrogel and thus not easily
accessible to immune cells. However, the purpose of this study is to test the effects
of the myoglobin as oxygen vectors. As we move to clinical applications, it is
possible to shift to native systems (i.e. human for human) or to engineer
"humanised" Mb variants, i.e. engineered to remove antigenic regions.

12. Are there any observations or thoughts about how oxygen delivery affects
graft/material angiogenesis/vascular integration over time?

Wildtype myoglobin serves as an oxygen reservoir within muscle tissue. It will
reversibly bind oxygen, extracting the molecule from the blood supply to
subsequently release it to the myocytes to maintain an oxygen tension of
approximately 2.5 torr (Garry & Mammen, 2007). This is especially important during
exercise when the metabolic demand of muscle tissue is high and there is risk of
hypoxia. In our application, we are investigating whether myoglobin will
dynamically bind and release oxygen to rescue the transplanted cells from hypoxia
during the critical early stages of development. Inevitably, as time goes on the
myoglobin-facilitated oxygen stores will deplete again, but ideally by this stage the
cells are more mature and angiogenesis/vascular integration has occurred. More
work is required to understand the dynamics of this process, but it is supported by
the difference in high and low affinity sperm whale myoglobin. The survival and
integration of grafts with high affinity myoglobin are improved, presumably
because the oxygen is withheld for longer as metabolic demand increases.

We have expanded on this within the text:

"...Wildtype myoglobin serves as an oxygen reservoir within muscle tissue. It will
reversibly bind oxygen, extracting the molecule from the blood supply to
subsequently release it to the myocytes to maintain an oxygen tension of
approximately 2.5 torr⁵¹. This is especially important during exercise when the
metabolic demand of muscle tissue is high and there is risk of hypoxia. Therefore,
we hypothesise that in our application the presence of Mb allows these hydrogels
to act as a temporary 'bloodstream' providing acute care for the grafted cells prior
to adequate angiogenesis and vascularisation within the graft. The release of
oxygen from myoglobin is dependent on the local partial pressure of oxygen within
the cell laden material. As such, the cellular respiration will inevitably deplete the

myoglobin-facilitated oxygen stores, but ideally by this stage the cells are more
mature and angiogenesis/ vascular integration has occurred..."

13. How long are the biomaterials expected to be capable of oxygen release in vivo?

It is difficult to estimate a time. The process is better thought of as an oxygen
dependent process. Once the oxygen dissolved within the hydrogel and
surrounding tissues is depleted below the affinity of the myoglobin variant, it will
release bound oxygen. Therefore, the low affinity myoglobin will release oxygen
earlier than the high affinity variant. Once oxygen is released from myoglobin and
consumed by the cells, it is depleted.

We have expanded on this within the text, as seen in our response to point 12
above.

Reviewer #3 (Remarks to the Author):

The manuscript by Wang et al. is aiming to solve the lack of oxygen within cell grafts
after transplantation into the brain and thus increase cell survival. The proposed solution
of using an engineered hybrid hydrogel, containing oxygen-binding myoglobin, is
innovative and interesting, and the results seem promising. The platform has the
potential be used for a variety of injuries and neurodegenerative conditions. We
recommend major revisions to address the following issues:

1. The rationale behind the groups "horse", "low affinity whale" etc. is only explained
during the last section of the results, and is quite confusing throughout the paper. Please
either move that section earlier in the results, or explain in the text the differences
between the groups and the figures legends.

Thank you for this valuable comment. Please see response to point 2, Reviewer 2.
We have restructured the manuscript to introduce all systems at the start and
explain how they differ.

2. In the results section, the authors mention: "... compliance matched to that of the
rodent brain ($G' \approx 100$ Pa, $G'' = 100$ Pa)". Please provide a reference or results to support
the mentioned values.

This is a great suggestion. The following reference has been added to support this
statement:

"...Janmey, P.A. and R.T. Miller, Mechanisms of mechanical signaling in
development and disease. Journal of Cell Science, 2011. 124(Pt 1): p. 9-18..."

3. Please better explain the results in supplementary figure 3 and add a legend. What indicates the functional state of the Mb, with and without the gel?

Thank you for bringing this to our attention, we have now updated Supplementary Figure 3 (now 5)..

The following has been added to line 205-209:

"...For the Mb to be functional, oxygen must be stabilised within the binding pocket by a ferrous iron (Fe²⁺) covalently bound to a heme prosthetic group. The change in the oxidation state of iron within Mb yields a characteristic absorbance spectrum, which provides an indication that the proteinaceous portion of Mb is stable and in a functional state as it successfully directs oxygen to the heme binding pocket.."

4. In the biocompatibility section:

a. To support the statement of "no influence on the immune system", please describe the microglial reaction to the hydrogel.

Thank you for the suggestion. Please see response to Reviewer 2, point 10. This
was a fantastic suggestion.

b. Please explain the experimental groups better: Did the "cell" group had no gel? Did
other groups all contain cells and gel? Please address this in the text as well.

Thank you for your suggestion. We have added the following:

"...Cell" group consisted only of transplanted GFP+ neural progenitor cells and
"gel" group included GFP+ neural progenitor cells embedded within Fmoc-
DDIKVAV hydrogel. All Mb groups, "Horse", "Whale", "Low affinity whale" and
"High affinity whale", included GFP+ neural progenitor cells, Fmoc-DDIKVAV
hydrogel and the respective Mb variant..."

c. Figure 2, legend, line #192 – where is image C? Please correct.

Thank you, this has been corrected.

5. "Sustained oxygen delivery" section:

a. The cells are called "cortical progenitor cells" (line # 198), yet they are being
transplanted into the striatum. Please explain the rationale.

Thank you for seeking clarification on this point. Placing cells into the thin mouse
cortex, without backflow of cells at the injection site is extremely difficult. To ensure
a robust comparison of the grafting conditions could be conducted, cells were
placed into the mouse parenchyma at a site that is both easy to target (due to its
large size) and cortical cells are known to robustly survive (noting that the striatum
is one of many targets for cortical projection neurons – i.e., the corticostriatal
pathway). We, and many others, commonly use the striatum as the implantation
site to test grafting conditions.

Please let us know if you think it is valuable to add this discussion into the
manuscript. We have not included it at this stage but will be happy to if the
reviewer thinks it is warranted.

b. To strengthen the results, it would be great to add high magnification images of the
GFP+ fibers that integrated into the host tissue (lines # 215-217).

Thank you for the suggestion. We have included GFP IHC-fluorescence images of
the surrounding host striatum as a supplementary figure (Supplementary Figure
S7) to complement the data. Due to the high sensitivity of fluorescence and low
number of GFP+ fibres present in the striatum across the groups, we believe that

fluorescent rather than high magnitude chromogenic images provide better
visualisation of the fibre innervation.

c. Lines 239-241 – the authors state that “Cell control and the IKVAV epitope-containing hydrogel resulted in no difference in NeuN number within the graft core”. It seems that this result contradicts earlier results by the group. Please add a possible explanation to the manuscript.

Thank you for bringing this to our attention. Although there is no statistical difference between the NeuN number of the cell only control and cell:hydrogel groups, there is a slight increase in the average which reflects trends observed in our previously reported work. For this proof-of-concept study, the group sizes we had used were relatively small compared with our previously reported results which would account for the smaller difference between groups. As such, we have amended the text to highlight this trend:

“...The IKVAV epitope-containing hydrogel resulted in a slight increase in the NeuN number (2801 ± 391 cells/graft) within the graft core compared with the cell only control group (2191 ± 273 cells/graft) (Figure 4A). This reflects trends we have observed in previously reported work...”

For this study, we wanted to highlight the significant differences observed between the cell:hydrogel group and those consisting of the myoglobin variants. These emphasise the importance of oxygen delivery to maintain homeostasis within the graft, in addition to the effect that the oxygen affinity of myoglobin has on cell survival, innervation and differentiation.

d. A better explanation on the difference between Fig. 4A and Fig. 4B would benefit the paper.

Thank you for your recommendation. We have added the following:

"...To investigate the effect of Mb on cell differentiation, we quantified the total
number of NeuN+ cells within each graft 28 days post transplantation (Figure 5A).
In light of the varying graft sizes observed (Figure 4A and C), we expressed the
total number of NeuN+ neurons as a proportion of total cells within the graft (i.e.
NeuN+GFP+/GFP+DAPI). We also assessed the percentage of NeuN+ cells
compared with undifferentiated GFP+DAPI+ cells (Figure 5B) to provide an
indication of the neuronal differentiation and density within the graft..."

We have also added the following to the description of Figure 4 (now Figure 5):

"...B: Percentage of NeuN+ cells in graft compared with undifferentiated
GFP+DAPI+ cells..."

e. Figure 4D – In some images (for example yellow) most cells co-localize (GFP+ and
NeuN+), whereas in others (the whale groups), most NeuN+ seems to be GFP-. Can you
determine whether those cells are transplanted cells, and not invading host cells?

Thank you. Please refer to our response to Reviewer 2, point 7.

f. Line 274 – Did the authors stain the tissue for endothelial markers, to determine which
cells express the Ki67? Otherwise, please rephrase.

We have previously shown that the Ki67+ proliferative cells present within human
pluripotent stem cell-derived neuronal grafts (matured over 9 months) were
predominantly host endothelial progenitors migrating into the graft. In this former
study, xenogeneic grafting of human cells into rats, and the use of antibodies
selective for rat endothelial progenitors (RECA-1) enable this clear conclusion. In
the present study, despite efforts to stain against the endothelial cells (using
CD31/PECAM), the staining (labelling mouse graft and mouse host) was poor
inconclusive. We appreciate the reviewers comment here and have therefore toned
down the conclusion here to suggest these ki67+ cells within the graft may be
endothelial progenitors and that further studies will be required to shed more light.

"...Further investigation is therefore required to determine whether the Ki67+ cells
presented in this study may be endothelial progenitor cells undergoing angiogenesis
within the graft tissue to form new blood vessels..."

6. When transplanting cells in the gels:

a. Were the cells characterized after isolation/before transplantation? Was cell viability
assayed before mixing with the gel? Please add those results.

Thank you. We had checked the cell viability the time of making the cell
preparation.

The following has been incorporated into the methods section
"...Following dissociation of the cortical tissue, cell viability was confirmed to be
>90% by trypan blue staining prior to resuspending the cells at a final density of
100,000 cells/ul'..."

b. Do injections of 2ul into the healthy brain tissue cause damage?

The cells are administered into the brain via a pulled glass capillary, tapered to an
external diameter of just 200um, sufficient to deliver the cells, yet cause minimal
damage to the brain. This glass capillary is coupled to a Hamilton syringe to control
the volume of delivery. This cell implantation method is described in detail in the
book chapter by Thompson LH, Parish CL. 'Transplantation of fetal midbrain
dopamine progenitors into a rodent model of Parkinson's disease' *Methods Mol*
*Biol.* (2013);1059:169-80. doi: 10.1007/978-1-62703-574-3.

c. Did the authors test for endotoxins in the gels and the Mb (produced in E.Coli)?

While the myoglobin was expressed heterologously in *E. coli*, we purified the
myoglobin using affinity chromatography and size exclusion chromatography,
which has been demonstrated to separate endotoxin from proteins (Petsch, 2000).
Consistent with this, we also see no indication of increased immune response due
to the presence of endotoxin (**Figure 3, Supplementary Figure S6**).

This has been added to the text

"...The two-step purification of proteins via affinity and size exclusion
chromatography, as performed here, has been shown to be sufficient to remove
endotoxin from protein preparations...."

34 d. Can the cells survive in the gels in vitro?

We have previously demonstrated that the 3D cultures involving Fmoc-DDIKVAV
SAP hydrogels provide a supportive and stable microenvironment for the
maintenance and long-term viability of neuronal stem cells after 7 days (Wang et
al., 2020) and astrocytes after 28 days (Maclean et al., 2018). For in vitro
applications and 3D culture studies, the cells are either seeded on top of,
underneath or within the hydrogel and are viable in all instances.

We have incorporated the following into the introduction section:

"...We have previously demonstrated that our self-assembling peptide (SAP)
hydrogels consisting of a bio-active epitope derived from laminin (IKVAV) promote
neuronal differentiation and integration *in vivo*¹⁻³. Furthermore, the ability of the
hydrogel to support neuronal stem cells⁴ and astrocytes⁵ for up to 7 and 28 days,
respectively, *in vitro* highlights the superior biocompatibility of the material..."

7. Last section of the results should be earlier in the paper to explain the rationale and
the experiments.

Please see response to point 2, Reviewer 2. We have restructured the manuscript to
introduce all systems at the start and explain how they differ. This is a great
suggestion by both reviewers and has significantly improve the manuscript.

a. When sacrificing the animals at 28 days, was the gel still present in the tissue?

Fmoc-SAPs consist of polypeptide chains of amino acids. These chemical units are
present within the body and thus processed via natural physiological mechanisms.
Furthermore, Fmoc-SAP hydrogels are formed through non-covalent bonds that
are weak and easy to disrupt (i.e. do not require chemical or enzymatic reactions).
It is therefore reasonable to hypothesise that our Fmoc-DDIKVAV SAP will degrade
*in vivo* over time but will not elicit a detrimental inflammatory response. We have
demonstrated this *in vivo* where we have shown the presence of the hydrogel in
the rodent brain 9 months after implantation (Soma et al., 2017).

b. For how long can Mb release oxygen?

Thanks for this suggestions. Please see our response to point 13, Reviewer 2.

8. In the discussion, lines # 394-397 – Did the authors look for neo-vascularization?

Thanks for this suggestion. Despite efforts to stain against the endothelial cells
(using a number of CD31/PECAM), the staining (labelling mouse graft and mouse
host) was poor and inconclusive. Further work is required to investigate the long-
term impact of oxygen delivery via myoglobin on vascularisation of the graft.

Would be interesting to see.

9. In the methods section:

a. Line # 436 – please add figure number.

Thank you, this has been corrected.

b. In vivo transplantation:

i. Were the gels prepared before mixing with the cells?

The cells were prepared prior to mixing with the cells. We have reworded this part
of the method to provide further clarity:

"...Hydrogels were prepared as previously stated and sterilized by UV lamp for 2
8 hours. Stock myoglobin was filtered by syringe filters (0.2 μm). Each gel underwent
mechanical sheering via vortexing and was mixed at a 1:1 ratio with cells to a final
gel and cells concentration of gel 10 mg mL^{-1} and 50,000 cells ul^{-1} , respectively,
immediately prior to in vivo delivery⁵⁶ ..."

ii. What was the final concentration of the gels?

As above, the following has been added:

"...Each gel underwent mechanical sheering via vortexing and was mixed at a 1:1
ratio with cells to a final gel and cells concentration of gel 10 mg mL^{-1} and 50,000
cells ul^{-1} , respectively, immediately prior to in vivo delivery⁵⁶..."

iii. How many cells were transplanted per animal?

As above, cells were prepared at 100,000cells/ul. The cells were then mixed at a 1:1
ratio with gel (or media) to give a final concentration of 50,000cells/ul. We then
injected 2ul/striatum i.e. 100,000 cells in total.

This has been added to line 609.

**References:**

- Bye, C. R., Thompson, L. H., & Parish, C. L. (2012). Birth dating of midbrain dopamine
neurons identifies A9 enriched tissue for transplantation into Parkinsonian mice.
*Experimental Neurology*, 236(1), 58–68.
<https://doi.org/10.1016/j.expneurol.2012.04.002>
- Chin, K., Khattak, S. F., Bhatia, S. R., & Roberts, S. C. (2008). Hydrogel-Perfluorocarbon
Composite Scaffold Promotes Oxygen Transport to Immobilized Cells.
*Biotechnology Progress*, 24(2), 358–366. <https://doi.org/10.1021/bp070160f>
- Farris, A. L., Rindone, A. N., & Grayson, W. L. (2016). Oxygen delivering biomaterials for
tissue engineering. *Journal of Materials Chemistry B*, 4(20), 3422–3432.
<https://doi.org/10.1039/C5TB02635K>
- Garry, D. J., & Mammen, P. P. A. (2007). Molecular Insights into the Functional Role of
Myoglobin. In R. C. Roach, P. D. Wagner, & P. H. Hackett (Eds.), *Hypoxia and the*
*Circulation* (pp. 181–193). Springer US.

Petsch, D. (2000). Endotoxin removal from protein solutions. *Journal of Biotechnology*,
76(2–3), 97–119. [https://doi.org/10.1016/S0168-1656\(99\)00185-6](https://doi.org/10.1016/S0168-1656(99)00185-6)
Somaa, F. A., Wang, T.-Y., Niclis, J. C., Bruggeman, K. F., Kauhausen, J. A., Guo, H.,
McDougall, S., Williams, R. J., Nisbet, D. R., Thompson, L. H., & Parish, C. L. (2017).
Peptide-Based Scaffolds Support Human Cortical Progenitor Graft Integration to
Reduce Atrophy and Promote Functional Repair in a Model of Stroke. *Cell Reports*,
20(8), 1964–1977. <https://doi.org/10.1016/j.celrep.2017.07.069>

Reviewer comments, second round review -

Reviewer #1 (Remarks to the Author):

The authors have generally improved the manuscript following receipt of reviewer comments and I am happy to recommend publication

Reviewer #2 (Remarks to the Author):

Thank you for thoroughly addressing the previous reviews. The manuscript is much improved.

Reviewer #3 (Remarks to the Author):

The authors have implemented the comments made by the reviewers and significantly improved the manuscript. We suggest accepting the manuscript.